# Comparison of blood and urine concentrations of equol by LC–MS/MS method and factors associated with equol production in 466 Japanese men and women

Remi Yoshikata[1,2], Khin Zay Yar Myint[1]*, Junichi Taguchi[1]

1 Tokyomidtown Medical Center, Minato-ku, Tokyo, Japan, 2 Hamasite Clinic, Minato-ku, Tokyo, Japan

* khinzayarmyint@gmail.com

**Data Availability Statement:** All relevant data are within the manuscript and its Supporting Information files.

## Abstract

Equol is produced from daidzein by the action of gut bacteria on soy isoflavones. However, not all people can produce equol, and metabolism differs even among the producers. We aimed to examine the equol producer status in both men and women, and investigate the relationships among the serum and urinary isoflavones as well as to other biomedical parameters. In this study, we measured the equol and daidzein concentrations from the blood and urine of 292 men and 174 women aged between 22 and 88 years by liquid chromatography-tandem mass spectrometry (LC–MS/MS). We then analysed the cut-off value for equol producers in both sexes, the relationship of serum and urinary equol concentrations, and other parameters, such as sex, age, endocrine function, glucose metabolism, lipid metabolism, and renal function with regards to equol-producing ability, among the different age groups. Equol producers were defined as those whose log ratio of urinary equol and daidzein concentration or log (equol/daidzein) was -1.42 or higher. Among 466 participants, 195 were equol producers (42%). The proportion of equol producers was larger in women. The cut-off value for equol producers was consistent in both sexes. Positive relationships were noted between serum and urinary equol levels in equol producers of both sexes; however, such a relationship was not detected in nonproducers. Lipid and uric acid abnormalities were more common with non equol producers in both men and women. Prostate specific antigen (PSA) levels in men were significantly lower in equol producers, especially in those in their 40 s. This study suggests a relationship between equol-producing ability and reduced risk of prostate disease as well as positive effects of equol on blood lipids and uric acid levels. However, lack of dietary information and disperse age groups were major drawbacks in generalizing the results of this study.

## Introduction

Dietary isoflavones are metabolized in the lower part of the small and large intestine into three main ingredients of soy isoflavones: daidzein, genistein and glycitein. There, the end-product or active metabolite of daidzein called 'equol' could be produced only in certain people who

**Funding:** Advanced Medical Care, Inc. and Himedic Medical Club gave the administrative support for this study but the funders had no role in study design, data collection and analysis, decision to publish, or preparation of the manuscript. The authors received no specific funding for this work.

**Competing interests:** The authors have declared that no competing interests exist.

have functional equol-producing bacteria [1]. Equol then enters the blood stream after being absorbed from the intestinal wall and is distributed to the target organs [2, 3]. Similar to other dietary end-products, it is chemically modified in the liver and mainly excreted in the urine, with only a small amount excreted into the stool [4]. It can be detected in the blood 8 hours after ingestion of isoflavones and reaches a peak concentration after 12 to 24 hours. After 72 hours, its blood level becomes negligible, as it is rarely reserved in the body [4].

Equol exists as enantiomers, R-equol and S-equol. However, S-equol is the only product that can be identified in the blood and urine of humans and animals [5, 6]. It has the highest potential for inducing health benefits among soy isoflavones, as it possesses estrogenic and antiestrogenic actions as well as antiandrogenic and antioxidant actions [7–9]. It is associated with the relief of menopausal symptoms and a reduced risk of related conditions, including osteoporosis [10–14], in women. For men, it was found to be associated with a reduced risk of prostate cancers [15, 16]. In addition, it was reported to have positive and antiaging effects on skin structures in both men and women [17, 18].

However, not everyone has the ability to produce equol [5]. The ability to produce equol depends on age, gender, genetics, dietary contents, and other factors [19–21]. Therefore, health benefits are not observed in some people even if they consume soy isoflavones. Even in those with equol-producing ability, individual and diurnal variations and the use of antibiotics greatly influence the desirable level of equol in the body [22–24]. For those reasons, there have been attempts to induce equol actions through supplements containing equol or with probiotics such as Lactobacilli and Bifidobacteria [25].

There are inconsistencies in the definitions of equol producers. For instance, some researchers used the minimum detectable levels of either serum or urinary equol [22, 26–28], which could vary with type of measurement. Some researchers defined equol producers with a cut-off value for either serum or urinary equol [29–32]. Some others recommended using the precursor and product ratio for accuracy. For example, Setchell (2006) et al. defined equol producers as those having a log10-transformed urinary equol to daidzein ratio of -1.75 and above after a soy challenge [33].

However, the above research findings were among the small group of people where soy challenge was relatively easier to implement. We need a robust definition to determine equol producers in larger populations where soy challenge is not feasible, such as those undergoing annual health screening. In 2018, Ideno et al. conducted an epidemiological study without soy challenge by measuring the urinary equol and daidzein levels of 4,412 Japanese women in the Japan Nurses' Health Study by the liquid chromatography-tandem mass spectrometry (LC-MS/CS) method [34]. There, they found out that those having a log10-transformed urinary equol to daidzein ratio of -1.42 and above can be classified as equol producers. However, their study was conducted only among women and there was no such study among men nor reproducibility of that definition has never been tested in both sexes. We hypothesized there were differences between men and women with regards to equol producing ability, i.e., they might have different cut-off values, as well as its relationship with other biomarkers inside the body. In this study, we aimed to examine cut-off value of equol producers, the relationship of blood and urinary equol, as well as their relationships with other parameters, such as sex, age, endocrine function, glucose metabolism, lipid metabolism, and renal function, in both sexes among the different age groups.

## Materials and methods

### Participants

This cross-sectional study was conducted among 466 healthy men and women (292 men and 174 women) aged between 22 and 88 (mean age 55) years who had undergone annual health

screening from June 2016 to December 2017 at the Himedic Kyoto University Hospital, a membership medical support facility where management and research are collaboratively carried out by Resorttrust, Inc and Kyoto University. All the participants hold the memberships of that facility. We were unable to obtain the information on the soy consumption habits or the last meals of the participants as they were not assessed in the ordinary health screening program.

## Ethical considerations

We included the data of the participants from all the men and women within the study period who were informed and provided a written general consent (S1 File) for the use of their health screening data for secondary purposes. The data were fully anonymized before they were assessed for research purpose on November 24, 2021. Although the study was conducted at the Kyoto University Hospital, we need to submit the proposal to a third-party ethical review board according to the regulations of Kyoto University Hospital for the use of secondary data by researchers from different affiliations. Therefore, the study was approved by the Institutional Review Board of The University of Tokyo.

## Isoflavone measurements

Equol and daidzein concentrations in the serum and urine were determined by the liquid chromatography/tandem mass spectrometry (LC-MS/MS) method, measured by LSI Medience Corporation (Tokyo, Japan). In brief, 100 μL of serum or 10-fold diluted urine was mixed with internal standards, followed by the addition of 150 μL of an β-glucuronidase enzyme solution for deconjugation (Roche Biochemical, Mannheim, Germany). Following a one-hour deconjugation reaction at 37˚C, free equol, daidzein, and genistein were purified using solid-phase extraction (Oasis PRiME HLB, Waters, Milford, MA). Subsequently, liquid chromatography (LC) -tandem mass spectrometry (LCMS-8050, Shimadzu, Japan) was employed with a reverse-phase LC column (ACQUITY UPLC HSS T3, 1.8 μm, 2.1 mm × 100 mm, Waters, Milford, MA) for analysis. Data processing was conducted using Mass Hunter software (Agilent, Santa Clara, CA). The peak areas were normalized using internal standards, and the concentration of each analyte was determined through a standard curve. The limits of detection (LODs) were 1 ng/mL for serum equol and daidzein, 10 ng/mL for urine equol, and 20.0 ng/mL for urine daidzein. Liquid chromatography–mass spectrometry (LC–MS) is an indispensable tool for quantitative and qualitative analysis in a wide range of fields, from pharmaceuticals and food science to environmental analysis. The advantage of LC-MS/MS over other methods such as glass chromatography-based methods for detection of isoflavones is that all the conjugated and unconjugated isoflavones and their metabolites can be separated and analysed faster and more efficiently [35, 36]. Therefore, it has been used extensively in quantitative measurements of isoflavones in several studies [37–41]. Based on a previous epidemiological study in 4,412 Japanese women [34], equol producers were defined as those whose log ratio of urinary equol and daidzein concentration or log (equol/daidzein) was -1.42 or higher.

## Metabolic parameters

Blood samples were obtained after an overnight fast to determine the fasting blood glucose level (FBG), hemoglobin A1c (HbA1c), glycated albumin, 1.5-Anhydro-D-glucitol, insulin, C-peptide, Homeostatic Model Assessment for Insulin Resistance (HOMA-IR), total cholesterol, low density lipoprotein cholesterol (LDL-C), triglycerides(TG), high density lipoprotein cholesterol (HDL-C), uric acid(UA), urinary creatinine(UCRE), high sensitivity C-reactive

protein (hs-CRP), estradiol (E2), luteinizing hormone (LH), follicle stimulating hormone (FSH), thyroid stimulating hormone (TSH), free-triiodothyronine (f-T3), free-thyroxine (f-T4).

### Statistical analysis

We first calculated the effected size for equol producer proportions by gender using Setchell's study on 23 men and 18 women, with equol producer proportions of 65% for men and 28% for women. Using that effect size (w = 0.162069), with the assumption of a two-sided 5% significance level and power of 80%, and calculated that a minimum of 415 subjects in total was necessary for the comparison of equol producers in gender by Chi-squared test. Considering 10% opt-out rate, we included data from 466 subjects. Serum and urinary S-equol and daidzein concentrations were expressed as micrograms/liter and when analysed were expressed as medians and interquartile ranges. The ratio of urinary equol to daidzein was calculated, transformed, and expressed as log10. We assessed the distributions of log10-transformed urinary equol: daidzein concentrations in histograms and used mixed models to examine any difference between the sexes to determine the equol producer status. According to the results, subjects with a ratio above −1.42 on this scale were classified as equol producers. We calculated the creatinine-corrected values of isoflavones to examine the associations between serum and urinary equol concentrations by linear regression analysis. The distribution of normality of the parameters was assessed with the Kolmogorov–Smirnov test, box plots, and histograms. We then compared the differences in parameters with regard to equol-producing ability using the Mann–Whitney U test. The proportions of abnormal values were compared using the chi-squared test. The cut-off values for each parameter were stated in the S3 File. All continuous variables are expressed as medians and interquartile ranges, and categorical variables are expressed as numbers and percentages. The calculations and Figure generation, except for the generation of the mixed model histograms, were performed using Microsoft Excel (Microsoft Corporation, 2019). The sample size calculations, and mixed model histograms were generated using R software (R 4.3.0, R Core Team, 2023). All tests were two-sided, and statistical significance was set to $p < 0.05$.

## Results

### Evaluation of equol producer status

We applied the same finite fixed model as the previous study to determine the optimal cut-off value to distinguish between equol producers and nonproducers [34]. The log10-transformed ratios of urinary equol to daidzein concentrations were plotted as shown in **Fig 1** across all participants (n = 466), as well as in men (n = 292) and women (n = 174) separately. In all distributions, we observed that the cut-off values were approximately -1.4, without significant distinction between men and women. Based on this definition, among 466 participants, 195 were equol producers (42%). The proportion of equol producers in women was 47%, whereas that in men was 39%.

### Comparisons of parameters between men and women participants

**Table 1** shows that women participants had higher equol concentrations in both serum and urine, glycated albumin, HDL-C, E2, LH, and FSH, which is consistent with other studies. Men had higher fasting blood glucose, 1.5-Anhydro-D-glucitol, C-peptide, HOMA-IR, TG, UA, UCRE, f-T3 and f-T4.

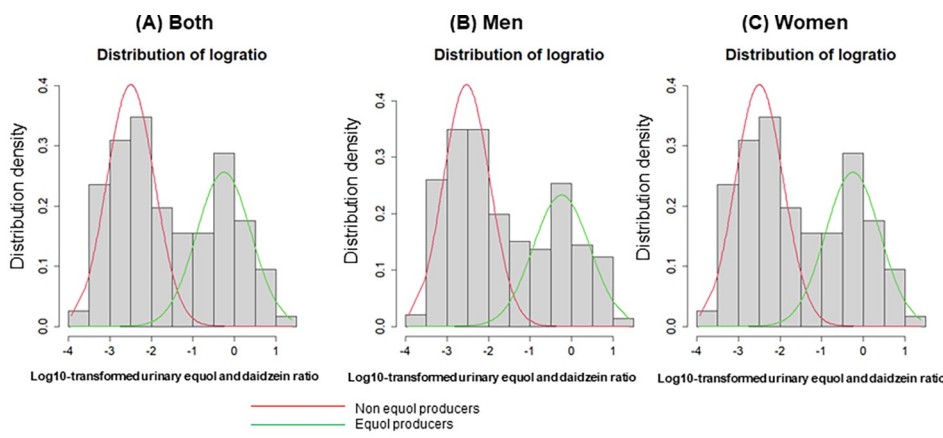

Determination of cut-off values in (A) all participants (N = 466), (B) males (N = 292), and (C) females (N = 174) using mixed model analysis. In all distributions, the cut-off values were approximately -1.4, without significant distinction between the male and female participants,

**Fig 1. Assessing the cut-off values for Log10-transformed urinary equol: Daidzein ratios.** Determination of cut-off values in (A) all participants (N = 466), (B) men (n = 292), and (C) women (n = 174) using mixed model analysis. In all distributions, the cut-off values were approximately -1.4, without significant distinction between the men and women, i.e., at the value of -1.42.

**Table 1. Comparisons of parameters between men and women participants.**

|  | Men (n = 292) |  | Women (n = 174) |  | p value |
| --- | --- | --- | --- | --- | --- |
| Equol producers | 114 | (39%) | 82 | (47%) | 0.136[a] |
| Age (years) | 54 | (22–86) | 54 | (28–88) | 0.377[a] |
| Serum equol (ng/dL) | 0.6 | (0.34, 3.37) | 1.03 | (0.61, 7.23) | <**0.001** |
| Serum daidzein (ng/dL) | 11.74 | (3.96, 35.35) | 17.85 | (7.16, 44.36) | **0.004** |
| Urinary equol (µg/gCr) | 7.52 | (3.7, 209) | 14.11 | (6.8, 600.8) | <**0.001** |
| Urinary daidzein (µg/gCr) | 1490.57 | (472.11, 4275.9) | 1876.56 | (761, 5997) | 0.07 |
| log (Equol/Daidzein) | -1.99 | (-2.74, 4275.9) | -1.59 | (-2.49, -0.3) | 0.121 |
| Estradiol (pg/mL) | <LOD | (0, 0) | 0.25 | (0.25, 66.55) | <**0.001** |
| Luteinizing hormone (mIU/mL) | <LOD | (0, 5.13) | 22.55 | (7.1, 32.68) | <**0.001** |
| Follicular stimulating hormone (mIU/mL) | <LOD | (0, 5.78) | 47.65 | (7.48, 70.23) | <**0.001** |
| C-reactive protein (mg/dL) | 0.05 | (0.05, 0.1) | 0.05 | (0.05, 0.1) | 0.167 |
| Thyroid stimulating hormone (µIU/mL) | 1.65 | (1.1, 2.51) | 1.83 | (1.21, 2.63) | 0.106 |
| Free thyroxine (ng/dL) | 1.28 | (1.18, 1.4) | 1.21 | (1.13, 1.32) | <**0.001** |
| Free triiodothyronine (pg/mL) | 3.14 | (2.88, 3.41) | 2.8 | (2.63, 3.06) | <**0.001** |
| Fasting blood glucose (mg/dL) | 94 | (88, 103) | 90 | (85.3, 95.75) | <**0.001** |
| Hemoglobin A1c (%) | 5.60 | (5.00, 6.00) | 5.70 | (5.00, 6.00) | 0.814 |
| Glycated albumin (%) | 1.30 | (1.30, 1.40) | 1.40 | (1.30, 1.50) | <**0.001** |
| 1.5-Anhydro-D-glucitol (µg/mL) | 18.7 | (13.48, 22.53) | 15.95 | (12.8, 19.48) | <**0.001** |
| Insulin (µU/mL) | 5 | (3.38, 7.60) | 4 | (2.70, 5.30) | 0.423 |
| C-peptide (ng/mL) | 1.62 | (1.16, 2.08) | 1.16 | (0.91, 1.59) | <**0.001** |
| HOMA-IR (Ratio) | 1.17 | (0.76, 1.85) | 0.91 | (0.60, 1.32) | <**0.001** |
| Insulin resistance, n (%) | 40 | (14%) | 14 | (8%) | 0.093[a] |
| Total cholesterol (mg/dL) | 197.5 | (175.8, 226) | 202 | (181, 223) | 0.289 |
| Low density lipoprotein cholesterol (mg/dL) | 115 | (99, 138) | 116.5 | (97, 137 | 0.878 |
| Triglycerides (mg/dL) | 114.5 | (74, 177.25) | 69.5 | (53.3, 106.5) | <**0.001** |
| High density lipoprotein cholesterol (mg/dL) | 55 | (47, 65) | 69 | (55.3, 78) | <**0.001** |
| Uric acid (mg/dL) | 6.15 | (5.4, 6.9) | 4.65 | (3.9, 5.38) | <**0.001** |

*(Continued)*

**Table 1.** (Continued)

|  | Men (n = 292) |  | Women (n = 174) |  | p value |
|---|---|---|---|---|---|
| Urinay creatinine (mg/dL) | 122 | (80.75, 173) | 72 | (50.3, 111.5) | **<0.001** |

Categorical variables were expressed as numbers (percentages). Continuous variables are expressed as medians (interquartile ranges) and were compared by the Mann–Whitney U test except for [a]: chi-squared test; statistically significant p values are bold. <LOD = lower than the limit of detection, HOMA-IR = Homeostatic Model Assessment for Insulin Resistance

### Relationship between serum and urine equol levels

Fig 2 showed strong positive relationships between the serum and urine equol levels in both men (r = 0.75, $R^2$ = 0.56, p<0.001) and women equol producers (r = 0.63, $R^2$ = 0.39, p<0.001). However, such a relationship was not observed in equol nonproducers in either sex (r = 0.24, $R^2$ = 0.0576, p<0.01, and r = 0.03, $R^2$ = 0.0008, p = 0.8, respectively). Equol nonproducers tended to have greater variances of urinary equol with reference to serum equol concentration, or they had almost no relationship between these two parameters. The relationship between serum and urinary equol concentrations was weaker in women equol producers, as their regression slope was lenient than that of men equol producers, although there were some individual differences.

### Comparison of other parameters between equol producers and nonproducers in men

Table 2 shows that PSA levels in men were significantly lower in equol producers (0.8 v.s. 1.0 ng/ml, p = 0.004), especially in men equol producers in their 40s (0.82 vs. 1.13 ng/ml, p<0.001) and 60 s (0.64 vs. 1.02 ng/ml, p<0.001), as shown in Fig 3A. In addition, a significant proportion of men with high LDL cholesterol levels were equol nonproducers (48.9% vs. 36.8%, p = 0.043), as shown in Fig 3B.

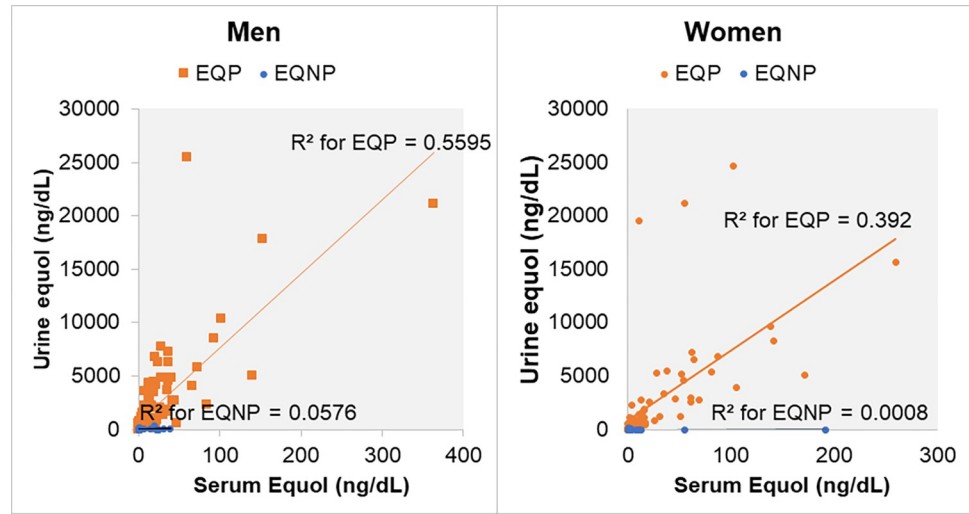

EQP=Equol producer, EQNP = equol non-producer

**Fig 2. Relationship between serum and urine equol levels in men and women participants.** Linear regression analysis in (A) men equol producers (n = 114) and nonproducers (n = 178), (B) women equol producers (n = 81) and nonproducers (n = 93). EQP = equol producers, EQNP = equol nonproducers.

**Table 2. Comparison between men equol producers and nonproducers.**

| | Equol producers (n = 114) | | Equol nonproducers (n = 178) | | p value |
|---|---|---|---|---|---|
| Age (years) | 54 | (45, 64) | 55 | (46, 64) | 0.6573 |
| Serum equol (ng/mL) | 7.64 | (0.64, 21.54) | 0.43 | (0.3, 0.72) | **<0.001** |
| Serum daidzein (ng/mL) | 8.25 | (2.97, 28.00) | 13.01 | (4.85, 36.84) | **0.025** |
| Urinary equol (μg/gCr) | 539.91 | (70.90, 2731.6) | 4.73 | (3.23, 8.23) | **<0.001** |
| Urinary daidzein (μg/gCr) | 875.25 | (151.73, 2808.75) | 2152.50 | (956.15, 5324.83) | **<0.001** |
| log (Equol/Daidzein) | -0.30 | (-0.64, 0.20) | -2.57 | (-2.98, -2.09) | **<0.001** |
| C-reactive protein (mg/dL) | 0.05 | (0.05, 0.1) | 0.05 | (0.05, 0.1) | 0.752 |
| Thyroid stimulating hormone (μIU/mL) | 1.51 | (1.06, 2.45) | 1.70 | (1.13, 2.53) | 0.366 |
| Free thyroxine (ng/dL) | 1.31 | (1.20, 1.4) | 1.27 | (1.17, 1.4) | 0.559 |
| Free triiodothyronine (pg/mL) | 3.16 | (2.89, 3.4) | 3.13 | (2.88, 3.42) | 0.907 |
| Fasting blood glucose (mg/dL) | 95.00 | (89.00, 103) | 94.00 | (88, 101.75) | 0.368 |
| Haemoglobin A1c (%) | 5.7 | (5.50, 6.00) | 5.6% | (5.00, 6.00) | 0.074 |
| Glycated albumin (%) | 1.3 | (1.30, 1.40) | 1.30 | (1.30, 1.40) | 0.929 |
| 1.5-Anhydro-D-glucitol (μg/mL) | 18.55 | (14.03, 22.90) | 18.85 | (13.13, 22.13) | 0.455 |
| Insulin (μU/mL) | 5.20 | (3.73, 7.78) | 4.90 | (3.13, 7.30) | 0.472 |
| C-peptide (ng/mL) | 1.65 | (1.23, 2.20) | 1.57 | (1.14, 2.00) | 0.215 |
| HOMA-IR (Ratio) | 1.23 | (0.85, 1.94) | 1.14 | (0.71, 1.81) | 0.355 |
| Insulin resistance, n (%) | 18 | (16%) | 22 | (12%) | 0.511 [a] |
| Total cholesterol (mg/dL) | 192.50 | (173.00, 225.75) | 200.00 | (179.25, 226.50) | 0.291 |
| Low density lipoprotein cholesterol (mg/dL) | 109.00 | (96, 134) | 119.00 | (100.25, 140.75) | 0.114 |
| Triglycerides (mg/dL) | 129.50 | (84.00, 187) | 107.50 | (72.25, 171.75) | 0.088 |
| High density lipoprotein cholesterol (mg/dL) | 55.00 | (46.00, 63.75) | 56.00 | (47.25, 65.75) | 0.204 |
| Uric acid (mg/dL) | 6.20 | (5.43, 6.90) | 6.10 | (5.40, 6.90) | 0.899 |
| Urinary creatinine (mg/dL) | 117.50 | (81.25, 174.50) | 125.00 | (80, 172.50) | 0.840 |
| Prostate specific antigen (ng/mL) | 0.80 | (0.34, 1.36) | 1.00 | (0.65, 1.53) | **0.004** |

Continuous variables are expressed as medians (interquartile ranges) and were compared by the Mann–Whitney U test except for [a]: chi-squared test; statistically significant p values are bold. HOMA-IR = Homeostatic Model Assessment for Insulin Resistance

### Comparison between equol producers and nonproducers in women

Among women, no significant quantitative differences were observed between equol nonproducers except for isoflavone parameters (**Table 3**). However, equol nonproducers had more abnormal LDL cholesterol, triglyceride, and uric acid levels (**Fig 4**).

## Discussion

We found that the cut-off value of the log-transformed urinary equol to daidzein ratio was almost consistent in both sexes, i.e., at the value of -1.42. Therefore, our definition of equol producers as those having a log 10-transformed ratio of urinary equol to daidzein of -1.42 or higher was relevant in our study population including both men and women. Previously, this cut-off value was reported in women participants only [34]. Therefore, this is the first study that could reproduce the same results in both men and women. Additionally, we found that urinary and serum equol concentrations were significantly correlated in equol producers but not in nonproducers. This also highlighted the important concept that it could be difficult to differentiate the equol producer phenotypes relying on either absolute serum or urinary equol concentrations. Therefore, Setchell et al proposed a robust method of defining an equol producer based upon the precursor-product relation using the log10-transformed ratio of urinary equol to its precursor daidzein [33].

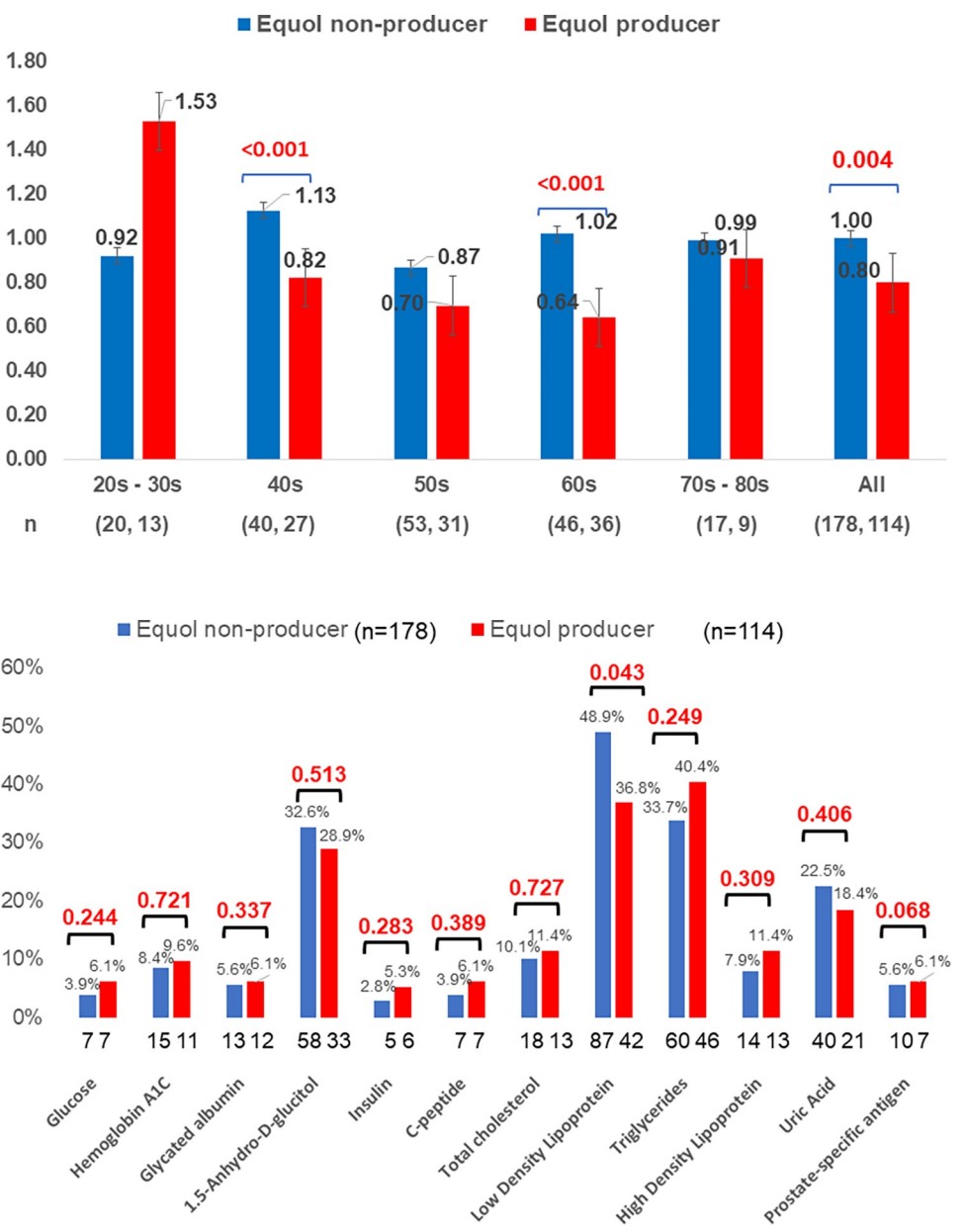

**Fig 3. Comparison of blood parameters between equol producers and nonproducers in men.** A: Comparisons of prostate-specific antigen levels between equol producers and nonproducers in each age group of men. PSA levels were expressed as medians and compared using the Mann–Whitney U test. B: Comparisons of abnormal metabolic values between equol producers and nonproducers in men. The proportions of abnormal values were compared using the chi-squared test.

The cut-off value of equol producers in Setchell's study was -1.75 after a standard 3-day challenge of soy foods containing isoflavones. Here, in our study, we used -1.42 as the cut-off value, as we did not perform the soy challenge, to reflect the real-time measurement as in the Japan Nurses' Health Study [34]. By using both the precursor and product relation to determine the equol producer phenotype, we also minimized errors due to large variance in equol concentrations due to differences in dietary isoflavone intake, pharmacokinetics, and methodologies for measuring the intrinsic equol. This definition identified 195 participants among

**Table 3. Comparison between equol producers and nonproducers in women.**

| | Equol producers (n = 82) | | Equol nonproducers (n = 92) | | p value |
|---|---|---|---|---|---|
| Age | 53.5 | (54, 60.5) | 55 | (46.25, 63) | 0. 8389 |
| Serum equol (ng/mL) | 6.77 | (1.06, 30.93) | 0.71 | (0.52, 1.14) | **<0.001** |
| Serum daidzein (ng/mL) | 13.67 | (5.73, 38.89) | 20.41 | (8.07, 51.81) | 0.1 |
| Urinary equol (μg/gCr) | 691.98 | (161.92, 2742.59) | 7.94 | (5.49, 12.20) | **<0.001** |
| Urinary daidzein (μg/gCr) | 1469.93 | (292.07, 3864.86) | 2907.89 | (1048.95, 7666.67) | **<0.01** |
| log (Equol/Daidzein) | -0.3 | (2.71, 3.54) | -2.46 | (2.79, 3.73) | **<0.001** |
| Estradiol (pg/mL) | 0.25 | (0.25, 51.60) | 0.25 | (0.25, 70.50) | 0.74 |
| Luteinizing hormone (mIU/mL) | 22.6 | (7.10, 35.20) | 22.55 | (6.60, 31.30) | 0.51 |
| Follicular stimulating hormone (mIU/mL) | 49.4 | (9.80, 76.20) | 47.65 | (6.30, 68.20) | 0.43 |
| C-reactive protein (mg/dL) | 0.05 | (0.05, 0.10) | 0.05 | (0.05, 0.10) | 0.57 |
| Thyroid stimulating hormone (μIU/mL) | 1.8 | (1.07, 2.79) | 1.83 | (1.33, 2.55) | 0.58 |
| Free thyroxine (ng/dL) | 1.22 | (1.14, 1.32) | 1.21 | (1.11, 1.31) | 0.3 |
| Free triiodothyronine (pg/mL) | 2.79 | (2.67, 3.09) | 2.8 | (2.59, 3.05) | 0.42 |
| Fasting blood glucose (mg/dL) | 90 | (86, 94) | 90 | (85, 97) | 0.64 |
| Haemoglobin A1c (%) | 5.6 | (5.4, 5.8) | 5.7 | (5.5, 6) | 0.05 |
| Glycated albumin (%) | 13.80 | (13.10, 14.70) | 14.00 | (14.00, 15.00) | 0.17 |
| 1.5-Anhydro-D-glucitol (μg/mL) | 15.90 | (13.50, 19.70) | 15.95 | (12.70, 19.40) | 0.55 |
| Insulin (μU/mL) | 4.10 | (3.00, 4.90) | 4.00 | (2.50, 5.80) | 0.98 |
| C-peptide (ng/mL) | 1.16 | (0.91, 1.58) | 1.16 | (0.92, 1.59) | 0.83 |
| HOMA-IR (Ratio) | 0.91 | (0.63, 1.16) | 0.91 | (0.55, 1.42) | 0.92 |
| Insulin resistance (%) | 6 | (7.40%) | 8 | (9%) | 0.99 [a] |
| Total cholesterol (mg/dL) | 201 | (183, 220) | 202 | (178, 224) | 0.86 |
| Low density lipoprotein cholesterol (mg/dL) | 115 | (98, 137) | 116.5 | (96, 137) | 0.61 |
| Triglycerides (mg/dL) | 70 | (55, 105) | 69.5 | (51, 110) | 0.84 |
| High density lipoprotein cholesterol (mg/dL) | 71 | (58, 80) | 69 | (55, 76) | 0.3 |
| Uric acid (mg/dL) | 4.6 | (4, 5.4) | 4.65 | (3.9, 5.3) | 0.9 |
| Urinary creatinine (mg/dL) | 71 | (51, 116) | 72 | (50, 100) | 0.63 |

Continuous variables are expressed as medians (interquartile ranges) and were compared by the Mann–Whitney U test except for [a]: chi-squared test; statistically significant p values are bold. HOMA-IR = Homeostatic Model Assessment for Insulin Resistance

466 as equol producers (42%), which was similar to the Japan Nurses' Health Study on 4,412 participants that has used the same cut-off value (41.5%). The proportion of equol producers in women was larger than that in men (47% versus 39%). The lenient nature of the relationship between serum and urinary equol levels in women might have reflected the stable serum equol concentration in women.

Studies on the benefits of soy isoflavones have yielded inconsistent results. This could be most likely due to the variations in equol producer phenotypes. Even in the studies that assessed the equol producer phenotypes, some results failed to reach statistical significance due to small sample sizes. For example, in this study, women equol nonproducers tended to have higher LDL cholesterol, triglyceride, high sensitivity C-reactive protein and uric acid levels, but the results were not statistically significant. However, in our previous study on 743 healthy women, equol producers in their 50s and 60 s, the age groups with declining estrogen levels, had favorable blood levels of lipids, uric acid, bone resorption markers, high sensitivity C-reactive protein, and homocysteine [42]. These positive effects were due to the estrogenic and antioxidant action of equol.

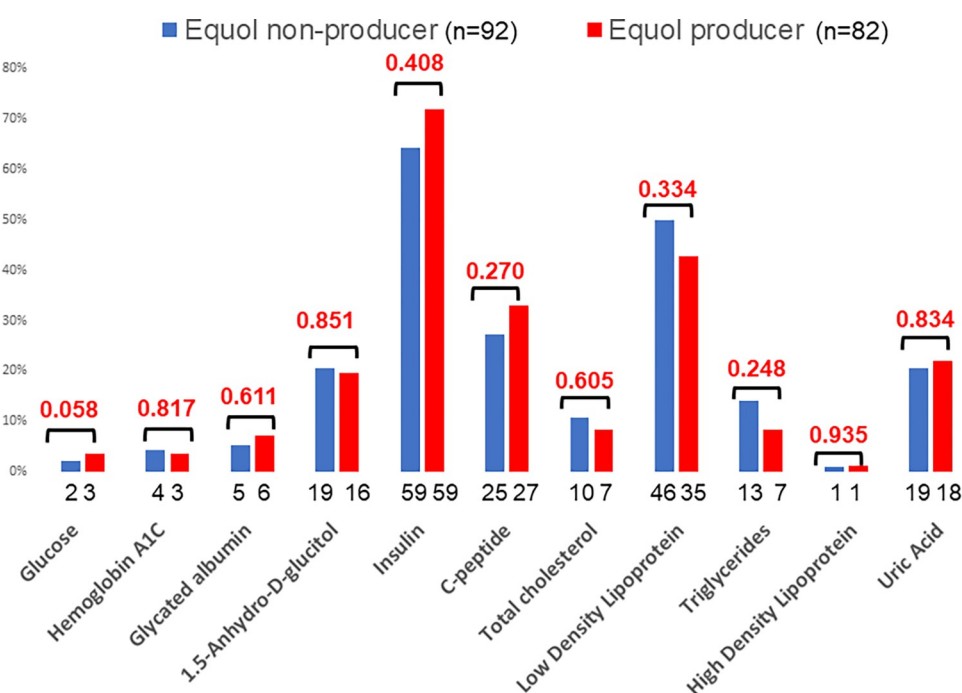

**Fig 4. Comparison of blood parameters between equol producers and nonproducers in women.** The proportions of abnormal values were compared using the chi-squared test.

In this study, men equol producers had significantly lower PSA levels than nonproducers, especially in men in their 40s and 60 s. Additionally, abnormal PSA values were rarely associated with men equol producers. Several studies have reported that equol can decrease serum PSA levels by its antiandrogenic action on 5-alpha-dehydro testosterone, decrease prostate size, and thus reduce the risk of prostate cancer [15, 16, 43, 44]. Therefore, this study has also added knowledge to equol benefits on men's prostate health. Additionally, it was a significant finding that abnormal LDL cholesterol levels were associated with men equol nonproducers. Furthermore, men equol nonproducers tended to have abnormal HDL cholesterol and uric acid levels, which needs to be explored in larger studies.

This is the first study to confirm the use of the precursor-product relation as a robust cut-off value to identify the equol producer phenotype in men and women using the LC–MS/MS method. It is also the first study that examined the differences between urine and serum equol concentrations in both women and men in equol producers and nonproducers. As equol producers were associated with better health benefits, findings in this study can be applied in integrating equol producer tests in the assessment of well-being for both sexes, especially in the middle age men and women with declining sex hormones. Determination of equol producers can also promote the healthy eating habits, and lifestyle promotion efforts [45].

However, this study has the following limitations. First, we did not have the detailed characteristics of the study participants, including medical history, anthropometric measures, and dietary habits. Therefore, that might affect the outcomes of the study. Especially the dietary habits might affect the proportion of equol producers since the isoflavone concentration in the body fluctuates with soy intake and is influenced by many other dietary factors. Second, the sample size varied among the different age groups. Age is one of the most important determinants affecting the parameters that we assessed. Therefore, we need to be cautious about generalizing the results in all age groups and need to consider further studies based on the sample

size calculations using the effect sizes in this study. Furthermore, the effect size we used in the sample size calculation was only based on the phenotype differences between genders and not applied for other outcomes. Especially, 37% difference between genders is not plausible. Lastly, as it is a cross-sectional study, causality cannot be determined.

## Conclusion

The proportion of equol producers was larger in women. The cut-off value of -1.42 for equol producers was consistent in both sexes. Positive relationships were noted between serum and urinary equol levels in equol producers of both sexes; however, such a relationship was not detected in nonproducers. The equol-producing ability tended to be higher in women, suggesting a relationship between estrogen and equol-producing ability. In men, equol-producing subjects had significantly lower PSA levels, suggesting a relationship between equol-producing ability and reduced risk of prostate disease. In addition, both women and men equol producers have positive effects on blood lipids and uric acid levels. However, we need more robust clinical trials in the representative samples of different age groups including dietary assessments to determine the health benefits of equol in both men and women.

## Supporting information

**S1 File. Consent form.**
(PDF)

**S2 File. Strobe checklist.**
(DOCX)

**S3 File. Cut-off values to determine abnormal results for each parameter.**
(TIF)

**S4 File. Minima data set.**
(XLSX)

## Acknowledgments

We would like to acknowledge all the participants who have consented to use their health screening data for the purpose of medical research, Himedic Kyoto University Hospital, and all the persons who have contributed to making this research possible.

## Author Contributions

**Conceptualization:** Remi Yoshikata, Khin Zay Yar Myint.

**Data curation:** Khin Zay Yar Myint.

**Formal analysis:** Khin Zay Yar Myint.

**Investigation:** Remi Yoshikata.

**Methodology:** Khin Zay Yar Myint.

**Project administration:** Remi Yoshikata, Junichi Taguchi.

**Resources:** Remi Yoshikata, Junichi Taguchi.

**Software:** Khin Zay Yar Myint.

**Supervision:** Junichi Taguchi.

**Validation:** Khin Zay Yar Myint.

**Visualization:** Khin Zay Yar Myint.

**Writing – original draft:** Remi Yoshikata.

**Writing – review & editing:** Khin Zay Yar Myint.

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
