## [Decision Letter · Decision Letter 0]

15 Dec 2023

PONE-D-23-20874Comparison of blood and urine concentrations of equol by LC‒MS/MS method and factors associated with equol production in 466 Japanese men and womenPLOS ONE

Dear Dr. Myint,

Thank you for submitting your manuscript to PLOS ONE. After careful consideration, we feel that it has merit but does not fully meet PLOS ONE’s publication criteria as it currently stands. Therefore, we invite you to submit a revised version of the manuscript that addresses the points raised during the review process.

When responding to reviewer comments, please make sure to clearly  detail your problem statement, introduction (with proper current citations), including thorough discussion of your results, which should also be properly written/presented. Your conclusion has to also reflect the findings/results form your study.

Please submit your revised manuscript by Jan 29 2024 11:59PM. If you will need more time than this to complete your revisions, please reply to this message or contact the journal office at plosone@plos.org. Please include the following items when submitting your revised manuscript:

We look forward to receiving your revised manuscript.

Kind regards,

Elingarami Sauli, PhD

Academic Editor

PLOS ONE

Journal Requirements:

"NO. The funders had no role in study design, data collection and analysis, decision to publish, or preparation of the manuscript."

Reviewers' comments:

Reviewer's Responses to Questions

**Comments to the Author**

1. Is the manuscript technically sound, and do the data support the conclusions?

Reviewer #1: Partly

Reviewer #2: Yes

Reviewer #3: No

2. Has the statistical analysis been performed appropriately and rigorously? 

Reviewer #1: No

Reviewer #2: Yes

Reviewer #3: No

3. Have the authors made all data underlying the findings in their manuscript fully available?

Reviewer #1: No

Reviewer #2: No

Reviewer #3: No

4. Is the manuscript presented in an intelligible fashion and written in standard English?

Reviewer #1: Yes

Reviewer #2: Yes

Reviewer #3: No

5. Review Comments to the Author

Reviewer #1: This cross-sectional study calculated a cutoff value for the benefit of equol-producers in Japanese men and women. While it is worth reporting that the cutoff was the same for both sexes (-1.42), the authors need to consider the following points.

Major points

1. The authors emphasize that there was a favorable trend in the equol producers, but this is not evident from the results or the figures. Therefore, the following corrections are needed, especially in Figure 3B and Figure 4:

・The cutoff values for each parameter should be stated (e.g., no cutoff value listed for High LDL cholesterol).

・List all P values as in the Table.

・Unify the description of the parameters in Table3B and Table4.

2. Conclusion: Please describe what you found out from this study (e.g., cutoff value of -1.42 for both men and women, association between blood and urine equol levels only in the producers, etc.), rather than describing the strengths of the study.

Minor points

3. Please include the abbreviation for PSA in the abstract.

4. Male/female and men/women are mixed in the paper. Please unify them.

5. Figure1: "Density" on the vertical axis should be changed to an appropriate term.

6. Table 1: Please include the abbreviation of the parameter in the footnote as well. Or, please provide the full name of the parameter in the table.

7. Table1：Please also indicate the median age.

8. Table2: Please indicate the median age of each of the producers and non-producers. Please adjust the decimal point of the parameter to Table3.

9. Table3: Please indicate the median age of each of the producers and non-producers.

Reviewer #2: Thank you very much for allowing me to review this important research undertaking. I have some comments:

Introduction:

1. The first paragraph only accounts for a single reference. I wonder if all those information were all lifted from the same source.

2. The introduction part is a bit weak. There should be a strong statement of the problem on the reasons why equol should be given priority and then transitioning to the research gap. Recent researches on epidemiological research on equol should also be stated to describe current landscape on such research topic.

Methods

1. Were the participants asked if they were soy drinkers and the frequency of drinking soy? This is an important variable as this might have an effect on the outcome measured.

Discussion:

Overall, the discussion is a weak. It should discuss public health implication of the findings and how can the government address the health issue on hand. Also, strengths of the study should also be included before the limitation part of the discussion.The limitation part also lacks in details in terms of the study design limitations. Kindly improve including the objectives.

Reviewer #3: In the study, urinary and blood isoflavones were determined in participants of annual health checkups. The associations between equol production and other various outcomes were investigated.

The findings on the relationship between blood and urinary levels are not surprising even though they may examine them in each sex and equol metabolic phenotype.

The cut-off value for equol phenotype was previously proposed by Ideno et al. and the authors in this study employed them and no additional investigation was conducted. Thus they found the threshold around -1.4 in log-scale and it does not give new insights.

The authors appeal that they found beneficial effects of equol on PSA in male population. In this study, the authors conducted comparisons of various outcomes other than PSA, and thus it is likely a statistical chance. Indeed, other outcomes showed statistically significant differences between equol phenotypes, but the authors focused only on PSA, this is cherry-picking and p-value hacking.

Background of participants is highly unclear.

During Jun 2016¬ to Dec 2017, only 466 persons visited the health screening? Total number of examined persons should be given. In addition, the participants were recruited at Kyoto University but the IRB approval was provided by University of Tokyo. Why?

Further, ranges of ages are too wide to evaluate the possible subclinical effects.

Sample size estimation was not appropriate. The effect size was only based on the phenotype differences between genders and not applied for other outcomes. Especially, 37% difference between genders is not plausible.

In addition, various outcomes examined in this study are affected by other background characteristics of participants. Absence of the information is critical to investigate potential relationship between equol phenotype and those outcomes.

Other points:

Why blood isoflavone levels should be corrected by creatinine?

Significant digits should be unified through the texts and tables.

The authors stated “However, it would not affect the results of this epidemiological study significantly”, but this does not make sense.

Figure 3A should be with error bars.

In Figure 3B and Figure 4, numbers in each category and p-values should be given.

6. PLOS authors have the option to publish the peer review history of their article (what does this mean?). If published, this will include your full peer review and any attached files.

Reviewer #1: No

Reviewer #2: No

Reviewer #3: No

---

## [Author Response · Author response to Decision Letter 0]

29 Jan 2024

Journal Requirements:

 Thank you for your references. We have revised the manuscript to meet the requirements.

 Thank you for your advice. We will deposit the raw data there when the manuscript was accepted.

 Thank you for your advice. We have added those points in the ethical consideration section.

 Thank you for your advice. We have added those points in the ethical consideration section.

 Thank you for your advice. We have added those points in the submission form.

"NO. The funders had no role in study design, data collection and analysis, decision to publish, or preparation of the manuscript."

 Thank you for your advice. We have added those points in the cover letter.

 Thank you for your advice. We will deposit the raw data there when the manuscript was accepted.

 Thank you for your advice. We have added the data set as Supporting information.

Reviewer's Responses to Questions

Comments to the Author

Reviewer #1: This cross-sectional study calculated a cutoff value for the benefit of equol-producers in Japanese men and women. While it is worth reporting that the cutoff was the same for both sexes (-1.42), the authors need to consider the following points.

Major points

1. The authors emphasize that there was a favorable trend in the equol producers, but this is not evident from the results or the figures. Therefore, the following corrections are needed, especially in Figure 3B and Figure 4:

・The cutoff values for each parameter should be stated (e.g., no cutoff value listed for High LDL cholesterol).

・List all P values as in the Table.

・Unify the description of the parameters in Table3B and Table4.

 Thank you for your valuable advice. The above points have been addressed in the figures and tables.

2. Conclusion: Please describe what you found out from this study (e.g., cutoff value of -1.42 for both men and women, association between blood and urine equol levels only in the producers, etc.), rather than describing the strengths of the study.

 Thank you for your insightful advice. The above points have been addressed in the conclusion.

Minor points

3. Please include the abbreviation for PSA in the abstract.

 Thank you for your insightful advice. The above points have been addressed in the abstract.

4. Male/female and men/women are mixed in the paper. Please unify them.

 Thank you for your insightful advice. The above points have been addressed.

5. Figure1: "Density" on the vertical axis should be changed to an appropriate term.

 Thank you for your advice. The above points have been addressed in the figure 1.

6. Table 1: Please include the abbreviation of the parameter in the footnote as well. Or, please provide the full name of the parameter in the table.

 Thank you for your insightful advice. The above points have been addressed in table 1.

7. Table1：Please also indicate the median age.

 Thank you for your insightful advice. The above points have been addressed in table 1.

8. Table2: Please indicate the median age of each of the producers and non-producers. Please adjust the decimal point of the parameter to Table3.

 Thank you for your insightful advice. The above points have been addressed in table 3. 

9. Table3: Please indicate the median age of each of the producers and non-producers.

 Thank you for your insightful advice. The above points have been addressed in table 3.

Reviewer #2: Thank you very much for allowing me to review this important research undertaking. I have some comments:

Introduction:

1. The first paragraph only accounts for a single reference. I wonder if all those information were all lifted from the same source.

 Thank you for your insightful advice. References were added in the first paragraph of the introduction. 

2. The introduction part is a bit weak. There should be a strong statement of the problem on the reasons why equol should be given priority and then transitioning to the research gap. Recent researches on epidemiological research on equol should also be stated to describe current landscape on such research topic.

 Thank you for your insightful advice. Epidemiological research on equol, statement of the problem and rationale for research objective have been added in the introduction. 

Methods

1. Were the participants asked if they were soy drinkers and the frequency of drinking soy? This is an important variable as this might have an effect on the outcome measured.

 Thank you for your insightful advice. We were not able to collect dietary assessments in this research and revised the methods and limitation sections for this.

Discussion:

Overall, the discussion is a weak. It should discuss public health implication of the findings and how can the government address the health issue on hand. Also, strengths of the study should also be included before the limitation part of the discussion.The limitation part also lacks in details in terms of the study design limitations. Kindly improve including the objectives.

 Thank you for your valuable advice. We have addressed the points in the discussion section.

Reviewer #3: In the study, urinary and blood isoflavones were determined in participants of annual health checkups. The associations between equol production and other various outcomes were investigated.

The findings on the relationship between blood and urinary levels are not surprising even though they may examine them in each sex and equol metabolic phenotype.

The cut-off value for equol phenotype was previously proposed by Ideno et al. and the authors in this study employed them and no additional investigation was conducted. Thus they found the threshold around -1.4 in log-scale and it does not give new insights.

The authors appeal that they found beneficial effects of equol on PSA in male population. In this study, the authors conducted comparisons of various outcomes other than PSA, and thus it is likely a statistical chance. Indeed, other outcomes showed statistically significant differences between equol phenotypes, but the authors focused only on PSA, this is cherry-picking and p-value hacking.

Background of participants is highly unclear.

During Jun 2016¬ to Dec 2017, only 466 persons visited the health screening? Total number of examined persons should be given. In addition, the participants were recruited at Kyoto University but the IRB approval was provided by University of Tokyo. Why?

Further, ranges of ages are too wide to evaluate the possible subclinical effects.

 Thank you for your advice. These participants were the members of the facility that we carried out research, not the entire hospital, therefore, the number is low. Although the study was conducted at the Kyoto University Hospital, we need to submit the proposal to a third-party ethical review board according to the regulations of Kyoto University Hospital for the use of secondary data by researchers from different affiliations. We have added those points in the method section. We have deleted the age ranges from the table as the numbers of subjects in each age range were too small if we change to narrower ranges of ages.

Sample size estimation was not appropriate. The effect size was only based on the phenotype differences between genders and not applied for other outcomes. Especially, 37% difference between genders is not plausible.

In addition, various outcomes examined in this study are affected by other background characteristics of participants. Absence of the information is critical to investigate potential relationship between equol phenotype and those outcomes.

 Thank you for your valuable advice. We have added those points as limitations of our study.

Other points:

Why blood isoflavone levels should be corrected by creatinine?

 Thank you for your valuable advice. We have corrected these levels.

Significant digits should be unified through the texts and tables.

 Thank you for your valuable advice. We have unified them.

The authors stated “However, it would not affect the results of this epidemiological study significantly”, but this does not make sense.

 Thank you for your valuable advice. We have added those points as limitations of our study.

Figure 3A should be with error bars.

 Thank you for your valuable advice. We have added error bars in figure 3A.

In Figure 3B and Figure 4, numbers in each category and p-values should be given.

 Thank you for your valuable advice. We have added those points in figure 3B and figure 4.

6. PLOS authors have the option to publish the peer review history of their article (what does this mean?). If published, this will include your full peer review and any attached files.

Do you want your identity to be public for this peer review? For information about this choice, including consent withdrawal, please see our Privacy Policy.

Reviewer #1: No

Reviewer #2: No

Reviewer #3: No

---

## [Decision Letter · Decision Letter 1]

13 Feb 2024

PONE-D-23-20874R1Comparison of blood and urine concentrations of equol by LC‒MS/MS method and factors associated with equol production in 466 Japanese men and womenPLOS ONE

Dear Dr. Myint,

Thank you for submitting your manuscript to PLOS ONE. After careful consideration, we feel that it has merit but does not fully meet PLOS ONE’s publication criteria as it currently stands. Therefore, we invite you to submit a revised version of the manuscript that addresses the points raised during the review process.

Be sure to include the following when submitting your responses to reviewer comments; proper labeling/naming of tables, figures, and concentrations. Also remember to include information on dietary habits, without forgetting proper discussion and conclusion of your findings/results.

We look forward to receiving your revised manuscript.

Kind regards,

Elingarami Sauli, PhD

Academic Editor

PLOS ONE

Journal Requirements:

Reviewers' comments:

Reviewer's Responses to Questions

**Comments to the Author**

1. If the authors have adequately addressed your comments raised in a previous round of review and you feel that this manuscript is now acceptable for publication, you may indicate that here to bypass the “Comments to the Author” section, enter your conflict of interest statement in the “Confidential to Editor” section, and submit your "Accept" recommendation.

Reviewer #3: (No Response)

Reviewer #4: All comments have been addressed

Reviewer #5: (No Response)

Reviewer #6: (No Response)

2. Is the manuscript technically sound, and do the data support the conclusions?

Reviewer #3: No

Reviewer #4: Yes

Reviewer #5: Yes

Reviewer #6: Partly

3. Has the statistical analysis been performed appropriately and rigorously? 

Reviewer #3: No

Reviewer #4: Yes

Reviewer #5: Yes

Reviewer #6: Yes

4. Have the authors made all data underlying the findings in their manuscript fully available?

Reviewer #3: Yes

Reviewer #4: Yes

Reviewer #5: Yes

Reviewer #6: Yes

5. Is the manuscript presented in an intelligible fashion and written in standard English?

Reviewer #3: Yes

Reviewer #4: Yes

Reviewer #5: Yes

Reviewer #6: No

6. Review Comments to the Author

Reviewer #3: Authors’ responses are not by point by point. I cannot understand what revisions were made to my comments.

Reviewer #4: Authors have appropriately addressed the comments and suggestions from the three reviewers in the first round of review. I have no more comment.

Reviewer #5: This study provides important information about isoflavone intake and health effects via the epidemiologic cross-sectional designed investigation. To report its achievements through Plos One, please consider and revise about comments below.

The conclusion should be included in one sentence in the abstract. You need a sentence like this. “This study suggests (or found) that ~ .”

Page 6. Please describe the abbreviation “LC/MS/MS” here (not page 9), because it was firstly used in this manuscript.

Several expressions on the page 12 are not good on the Results section. Expressions such as “consistent with our definition”, “relevant in our population” must be moved to the Discussion section.

Page 15. 1st sentence. observation → relationship

Figure 3A. Numbers of each age range are too small. I recommend unifying 20s and 30s as well as 70s and 80s.

Page 23. The sentences “In addition, ~ examined in this study.” are a duplication with the first limitation.

Authors emphasize LC/MS/MS method such an extent to include the title and key words. Then please explain the meaning of this method in your study. It is not a newly developed or validated method in this study, and readers may be curious about this method whether any special strength or not than other methods (i.e. GC-MS or anything else).

Page 24. 3rd line. Association → relationship

Reviewer #6: The article describes the investigation of the equol level in blood and urine samples from 466 Japanese men and women. Unfortunately, no information about the dietary habits are available which limits the suitability of this study. In my opinion it is important to state this limitation already in the abstract.

Cut-off value: I am sorry, but I do not understand how the authors “analysed” the cut-off value. As far as I understood it, they just took it from a previous publication. Please clarify it.

Furthermore, I have a problem with the statements of the authors drawing conclusions with the health data. First of all, there are no information about the dietary habits of the participants, so whether or not equol was detected in blood or urine greatly depends on the last consumption of isoflavone containing food. Secondly, the number of participants in the different age groups were rather limited. In my opinion, it is important to point out these limitations already in the abstract.

The authors provided the minimal data set, but without an explanation, it is not clear to me what the individual

columns are standing for.

#Please check the English language in the whole manuscript, e.g., sometimes you use articles where there should be no article (page 12, line 8 – there should not be an article before “between” “men and women”).

Keywords: As far as I have learnt it the key words should be different than the words used in the title.

Page 6 at the end of the page: What do you mean with “LC/MS/CS”? Do you mean “LC-MS/MS”?

Page 7 – line 2: I have learnt that “but” should not be used at the beginning of the sentence. Please consider using “However” instead.

Page 9: I am missing some details on the used method. Ideally you should provide them in the manuscript itself, but at least you should provide a reference.

Page 9 – metabolic parameters: Please check, but there should always be a space before the brackets (e.g., “glucose level (FBG)”). Furthermore, in English the compound names should be written with small starting letters (e.g., estradiol). Please check this in the whole manuscript.

Page 12 – evaluation of equol producer status: You state “we applied the same finite fixed model as the previous study”, but you do not provide a reference. Please add it here.

Page 12 – line 6: Please check, but you sometimes used “n” and sometimes “N”, please be consistent.

Table 1-3: Why did you change the order in these tables? I would use always the same order since these tables state the same information, but in different groups (men & women, men and women).

In Table 1: Why did you report serum concentrations as µg/g Cr? This unit belongs to the urinary concentrations, but not to serum. In the other two tables (2 and 3) you used the unit “ng/dL”. Furthermore, please never report analytical results of “zero” as in case of estradiol. Always state “lower than the limit of detection < LOD).

Please explain all used abbreviations – also “BMI” (page 23).

Supporting information file 3: Please check the HDL-cholesterol level – is there really a 10 fold difference between men and women? All other values are exactly the same between men and women

References: Please check the references carefully.

“in vitro”, “in vivo” should be written in general written in italics.

Reference 5: It should be a “beta (β)” symbol not a “ß” which is a kind of “s” in German.

Several times not only the year, but also the month of publication is provided. I think that it is not necessary to state the month as well.

Reference 11: Please check – I am not sure what the number eight (8) means prior to the author names.

Reference 43: I think that here something is missing. Please check.

Figures and Tables should be understandable without reading the manuscript itself. Therefore, please explain the used abbreviations (e.g., “EQP” and “EQNP” in Figure 2).

Figure 1: You changed the nomenclature in the whole manuscript to “men” and “women”; but in this figure you still you “male” and “female”. Please correct it here as well.

Figure 2: Please state the units reported in theses graphs. It is not clear to me what you did. Furthermore, in each subgraph for male and female two Rsquare values are provided, but only in case of male it is specified to which group it belongs. Moreover, you changed the nomenclature in the whole manuscript to “men” and “women”; but in this figure you still you “male” and “female”. Please correct it here as well.

Figure 3A Please check the numbers provided below the figure. In case of “20s” should it not be (0, 2) instead of (2, 0). As far as I understood it the first number belongs to equol non-producers and the second number to equal producers. Furthermore, in case of “80s”, how can you provide a standard deviation when only one person belonged to the group. Please also specify which type of standard deviation is provided.

Please unify, because in Figure 3B and 4 first equol-non-producers are provided, whereas in Figure 3A you changed the order and provide first equol producers and then equol-non-producers. In my opinion it would be less confusing if the order is always the same.

7. PLOS authors have the option to publish the peer review history of their article (what does this mean?). If published, this will include your full peer review and any attached files.

Reviewer #3: No

Reviewer #4: No

Reviewer #5: **Yes: **Yong Min Cho

Reviewer #6: No

---

## [Author Response · Author response to Decision Letter 1]

28 Feb 2024

Reviewer #3: Authors’ responses are not by point by point. I cannot understand what revisions were made to my comments.

We apologize for not addressing point by point. Let us respond your previous comments as follows. 

Responses for the previous comments by Reviewer #3.

Reviewer #3: In the study, urinary and blood isoflavones were determined in participants of annual health checkups. The associations between equol production and other various outcomes were investigated.

The findings on the relationship between blood and urinary levels are not surprising even though they may examine them in each sex and equol metabolic phenotype.The cut-off value for equol phenotype was previously proposed by Ideno et al. and the authors in this study employed them and no additional investigation was conducted. Thus, they found the threshold around -1.4 in log-scale and it does not give new insights.

Thank you for your insightful comments. We have mentioned the distinction in the discussion as follows.

" Previously, this cut-off value was reported in women participants only [34]. Therefore, this is the first study that could reproduce the same results in both men and women. Additionally, we found that urinary and serum equol concentrations were significantly correlated in equol producers but not in nonproducers. This also highlighted the important concept that it could be difficult to differentiate the equol producer phenotypes relying on either absolute serum or urinary equol concentrations."

The authors appeal that they found beneficial effects of equol on PSA in male population. In this study, the authors conducted comparisons of various outcomes other than PSA, and thus it is likely a statistical chance. Indeed, other outcomes showed statistically significant differences between equol phenotypes, but the authors focused only on PSA, this is cherry-picking and p-value hacking.

Thank you for your comment. We have discussed other outcomes as follows.

" Studies on the benefits of soy isoflavones have yielded inconsistent results. This could be most likely due to the variations in equol producer phenotypes. Even in the studies that assessed the equol producer phenotypes, some results failed to reach statistical significance due to small sample sizes. For example, in this study, women equol nonproducers tended to have higher LDL cholesterol, triglyceride, high sensitivity C-reactive protein and uric acid levels, but the results were not statistically significant. However, in our previous study on 743 healthy women, equol producers in their 50s and 60 s, the age groups with declining estrogen levels, had favorable blood levels of lipids, uric acid, bone resorption markers, high sensitivity C-reactive protein, and homocysteine [40]. These positive effects were due to the estrogenic and antioxidant action of equol."

Background of participants is highly unclear.

During Jun 2016¬ to Dec 2017, only 466 persons visited the health screening? Total number of examined persons should be given. In addition, the participants were recruited at Kyoto University but the IRB approval was provided by University of Tokyo. Why?

Thank you for your advice. These participants were the members of the facility that we carried out research, not the entire hospital, therefore, the number is low. Although the study was conducted at the Kyoto University Hospital, we need to submit the proposal to a third-party ethical review board according to the regulations of Kyoto University Hospital for the use of secondary data by researchers from different affiliations. We have added those points in the method section as follows.

"Although the study was conducted at the Kyoto University Hospital, we need to submit the proposal to a third-party ethical review board according to the regulations of Kyoto University Hospital for the use of secondary data by researchers from different affiliations. Therefore, the study was approved by the Institutional Review Board of The University of Tokyo (Supplementary data file 2)."

Further, ranges of ages are too wide to evaluate the possible subclinical effects.

Thank you for your comment. We have deleted those age ranges in the tables 1-3.

Sample size estimation was not appropriate. The effect size was only based on the phenotype differences between genders and not applied for other outcomes. Especially, 37% difference between genders is not plausible.

In addition, various outcomes examined in this study are affected by other background characteristics of participants. Absence of the information is critical to investigate potential relationship between equol phenotype and those outcomes.

Thank you for your valuable advice. We have added those points as limitations of our study as follows.

" Furthermore, the effect size we used in the sample size calculation was only based on the phenotype differences between genders and not applied for other outcomes. Especially, 37% difference between genders is not plausible."

Other points:

Why blood isoflavone levels should be corrected by creatinine?

Thank you for your valuable advice. We have corrected these levels in the tables 1-3.

Significant digits should be unified through the texts and tables.

Thank you for your valuable advice. We have unified them.

The authors stated “However, it would not affect the results of this epidemiological study significantly”, but this does not make sense.

Thank you for your valuable advice. We have added those points as limitations of our study as follows.

"First, we did not have the detailed characteristics of the study participants, including medical history, anthropometric measures, and dietary habits. Therefore, that might affect the outcomes of the study. Especially the dietary habits might affect the proportion of equol producers since the isoflavone concentration in the body fluctuates with soy intake and is influenced by many other dietary factors."

Figure 3A should be with error bars.

Thank you for your valuable advice. We have added error bars in figure 3A.

In Figure 3B and Figure 4, numbers in each category and p-values should be given.

Thank you for your valuable advice. We have added those points in figure 3B and figure 4.

Reviewer #4: Authors have appropriately addressed the comments and suggestions from the three reviewers in the first round of review. I have no more comment.

Thank you for your comment. We really appreciate your valuable insights and suggestions.

Reviewer #5: This study provides important information about isoflavone intake and health effects via the epidemiologic cross-sectional designed investigation. To report its achievements through Plos One, please consider and revise about comments below.

The conclusion should be included in one sentence in the abstract. You need a sentence like this. “This study suggests (or found) that ~ .”

Thank you for your comment. We added that sentence in the abstract as follows.

"This study suggests a relationship between equol-producing ability and reduced risk of prostate disease as well as positive effects of equol on blood lipids and uric acid levels."

Page 6. Please describe the abbreviation “LC/MS/MS” here (not page 9), because it was firstly used in this manuscript.

Thank you for your comment. We described the abbreviation in page 6 as you suggested.

Several expressions on the page 12 are not good on the Results section. Expressions such as “consistent with our definition”, “relevant in our population” must be moved to the Discussion section.

Thank you for your comment. We moved that section to the Discussion section as you suggested.

Page 15. 1st sentence. observation → relationship

Thank you for your comment. We corrected the term as you suggested.

Figure 3A. Numbers of each age range are too small. I recommend unifying 20s and 30s as well as 70s and 80s.

Thank you for your comment. We have combined the age ranges of 30s and 40s and 70s and 80s and revised figure 3A.

Page 23. The sentences “In addition, ~ examined in this study.” are a duplication with the first limitation.

Thank you for your comment. We have deleted that sentence from limitation.

Authors emphasize LC/MS/MS method such an extent to include the title and key words. Then please explain the meaning of this method in your study. It is not a newly developed or validated method in this study, and readers may be curious about this method whether any special strength or not than other methods (i.e. GC-MS or anything else).

Thank you for your comment. We added those facts in the method section as follows.

"In brief, 100 µL of serum or 10-fold diluted urine was mixed with internal standards, followed by the addition of 150 µL of an β-glucuronidase enzyme solution for deconjugation (Roche Biochemical, Mannheim, Germany). Following a one-hour deconjugation reaction at 37°C, free equol, daidzein, and genistein were purified using solid-phase extraction (Oasis PRiME HLB, Waters, Milford, MA). Subsequently, liquid chromatography (LC) -tandem mass spectrometry (LCMS-8050, Shimadzu, Japan) was employed with a reverse-phase LC column (ACQUITY UPLC HSS T3, 1.8 µm, 2.1 mm × 100 mm, Waters, Milford, MA) for analysis. Data processing was conducted using Mass Hunter software (Agilent, Santa Clara, CA). The peak areas were normalized using internal standards, and the concentration of each analyte was determined through a standard curve."

"The advantage of LC-MS/MS over other methods such as glass chromatography-based methods for detection of isoflavones is that all the conjugated and unconjugated isoflavones and their metabolites can be separated and analysed faster and more efficiently. [35, 36]. Therefore, it has been used extensively in quantitative measurements of isoflavones in several studies　[37-41]."

Page 24. 3rd line. Association → relationship

Thank you for your comment. We corrected the term as you suggested.

Reviewer #6: The article describes the investigation of the equol level in blood and urine samples from 466 Japanese men and women. Unfortunately, no information about the dietary habits are available which limits the suitability of this study. In my opinion it is important to state this limitation already in the abstract.

Thank you for your comment. We added that sentence in the abstract as follows.

" However, lack of dietary information and disperse age groups were major drawbacks in generalizing the results of this study."

Cut-off value: I am sorry, but I do not understand how the authors “analysed” the cut-off value. As far as I understood it, they just took it from a previous publication. Please clarify it.

Thank you for your comment. We added the following points for what is unknown from the previous study and our hypothesis in the introduction before the objective of the study as follows.

" However, their study was conducted only among women and there was no such study among men nor reproducibility of that definition has never been tested in both sexes. We hypothesized there were differences between men and women with regards to equol producing ability, i.e., they might have different cut-off values, as well as its relationship with other biomarkers inside the body."

Furthermore, I have a problem with the statements of the authors drawing conclusions with the health data. First of all, there are no information about the dietary habits of the participants, so whether or not equol was detected in blood or urine greatly depends on the last consumption of isoflavone containing food. Secondly, the number of participants in the different age groups were rather limited. In my opinion, it is important to point out these limitations already in the abstract.

Thank you for your comment. We have added that sentence in the abstract and conclusion section as follows.

" However, lack of dietary information and disperse age groups were major drawbacks in generalizing the results of this study."

"However, we need more robust clinical trials in the representative samples of different age groups including dietary assessments to determine the health benefits of equol in both men and women."

The authors provided the minimal data set, but without an explanation, it is not clear to me what the individual columns are standing for.

Thank you for your comment. We have added the explanation of the columns in the minimal data set in a new sheet.

#Please check the English language in the whole manuscript, e.g., sometimes you use articles where there should be no article (page 12, line 8 – there should not be an article before “between” “men and women”).

Thank you for your comment. We have corrected that point as you suggested. 

Keywords: As far as I have learnt it the key words should be different than the words used in the title.

Thank you for your comment. We have changed the key words as you suggested. 

Page 6 at the end of the page: What do you mean with “LC/MS/CS”? Do you mean “LC-MS/MS”?

Thank you for your comment. We have corrected the abbreviation as you suggested. 

Page 7 – line 2: I have learnt that “but” should not be used at the beginning of the sentence. Please consider using “However” instead.

Thank you for your comment. We have corrected the vocabulary as you suggested. 

Page 9: I am missing some details on the used method. Ideally you should provide them in the manuscript itself, but at least you should provide a reference.

Thank you for your comment. We added those facts and references in the method section as follows.

"In brief, 100 µL of serum or 10-fold diluted urine was mixed with internal standards, followed by the addition of 150 µL of an β-glucuronidase enzyme solution for deconjugation (Roche Biochemical, Mannheim, Germany). Following a one-hour deconjugation reaction at 37°C, free equol, daidzein, and genistein were purified using solid-phase extraction (Oasis PRiME HLB, Waters, Milford, MA). Subsequently, liquid chromatography (LC) -tandem mass spectrometry (LCMS-8050, Shimadzu, Japan) was employed with a reverse-phase LC column (ACQUITY UPLC HSS T3, 1.8 µm, 2.1 mm × 100 mm, Waters, Milford, MA) for analysis. Data processing was conducted using Mass Hunter software (Agilent, Santa Clara, CA). The peak areas were normalized using internal standards, and the concentration of each analyte was determined through a standard curve."

"The advantage of LC-MS/MS over other methods such as glass chromatography-based methods for detection of isoflavones is that all the conjugated and unconjugated isoflavones and their metabolites can be separated and analysed faster and more efficiently. [35, 36]. Therefore, it has been used extensively in quantitative measurements of isoflavones in several studies　[37-41]."

Page 9 – metabolic parameters: Please check, but there should always be a space before the brackets (e.g., “glucose level (FBG)”). Furthermore, in English the compound names should be written with small starting letters (e.g., estradiol). Please check this in the whole manuscript.

Thank you for your comment. We have corrected those vocabularies as you suggested. 

Page 12 – evaluation of equol producer status: You state “we applied the same finite fixed model as the previous study”, but you do not provide a reference. Please add it here.

Thank you for your comment. We have added the reference as you suggested. 

Page 12 – line 6: Please check, but you sometimes used “n” and sometimes “N”, please be consistent.

Thank you for your comment. We have made "n" consistent as you suggested. 

Table 1-3: Why did you change the order in these tables? I would use always the same order since these tables state the same information, but in different groups (men & women, men and women).

Thank you for your comment. We have unified the order of the tables as you suggested. 

In Table 1: Why did you report serum concentrations as µg/g Cr? This unit belongs to the urinary concentrations, but not to serum. In the other two tables (2 and 3) you used the unit “ng/dL”. Furthermore, please never report analytical results of “zero” as in case of estradiol. Always state “lower than the limit of detection < LOD).

Thank you for your comment. We have corrected the units and used LOD in place of zero as you suggested. 

Please explain all used abbreviations – also “BMI” (page 23).

Thank you for your comment. We have explained the abbreviations as you suggested. 

Supporting information file 3: Please check the HDL-cholesterol level – is there really a 10 fold difference between men and women? All other values are exactly the same between men and women

Thank you for your comment. We have corrected the value as you pointed out. 

References: Please check the references carefully.

“in vitro”, “in vivo” should be written in general written in italics.

Thank you for your comment. We have written them in italics as you suggested. 

Reference 5: It should be a “beta (β)” symbol not a “ß” which is a kind of “s” in German.

Thank you for your comment. We have corrected the symbol as you suggested. 

Several times not only the year, but also the month of publication is provided. I think that it is not necessary to state the month as well.

Thank you for your comment. We have omitted the months as you suggested. 

Reference 11: Please check – I am not sure what the number eight (8) means prior to the author names.

Thank you for your comment. We have corrected that point as you suggested. 

Reference 43: I think that here something is missing. Please check.

Thank you for your comment. We have added some more information as you suggested. 

Figures and Tables should be understandable without reading the manuscript itself. Therefore, please explain the used abbreviations (e.g., “EQP” and “EQNP” in Figure 2).

Figure 1: You changed the nomenclature in the whole manuscript to “men” and “women”; but in this figure you still you “male” and “female”. Please correct it here as well.

Thank you for your comment. We have corrected that point as you suggested. 

Figure 2: Please state the units reported in theses graphs. It is not clear to me what you did. Furthermore, in each subgraph for male and female two Rsquare values are provided, but only in case of male it is specified to which group it belongs. Moreover, you changed the nomenclature in the whole manuscript to “men” and “women”; but in this figure you still you “male” and “female”. Please correct it here as well.

Thank you for your comment. We have corrected the points as you suggested. 

Figure 3A Please check the numbers provided below the figure. In case of “20s” should it not be (0, 2) instead of (2, 0). As far as I understood it the first number belongs to equol non-producers and the second number to equal producers. Furthermore, in case of “80s”, how can you provide a standard deviation when only one person belonged to the group. Please also specify which type of standard deviation is provided.

Thank you for your comment. We have corrected the points as you suggested and combined 20s and 30s, as well as 70s and 80s as the numbers are quite low.

Please unify, because in Figure 3B and 4 first equol-non-producers are provided, whereas in Figure 3A you changed the order and provide first equol producers and then equol-non-producers. In my opinion it would be less confusing if the order is always the same.

Thank you for your comment. We have corrected the points as you suggested.

---

## [Editor Report · Decision Letter 2]

1 Mar 2024

Comparison of blood and urine concentrations of equol by LC‒MS/MS method and factors associated with equol production in 466 Japanese men and women

PONE-D-23-20874R2

Dear Dr. Myint,

We’re pleased to inform you that your manuscript has been judged scientifically suitable for publication and will be formally accepted for publication once it meets all outstanding technical requirements.

Kind regards,

Elingarami Sauli, PhD

Academic Editor

PLOS ONE
---

## [Editor Report · Acceptance letter]

13 Mar 2024

PONE-D-23-20874R2 

PLOS ONE

Dear Dr. Myint, 

I'm pleased to inform you that your manuscript has been deemed suitable for publication in PLOS ONE. Congratulations! Your manuscript is now being handed over to our production team.

Kind regards, 

on behalf of

Dr. Elingarami Sauli 

Academic Editor

PLOS ONE